# Factors Shaping A/W Heat Pumps CO$_2$ Emissions—Evidence from Poland

**Piotr Jadwiszczak** [1,*], **Jakub Jurasz** [1], **Bartosz Kaźmierczak** [1], **Elżbieta Niemierka** [1,*] **and Wandong Zheng** [2]

1   Faculty of Environmental Engineering, Wrocław University of Science and Technology,
    PL50377 Wrocław, Poland; jakubkamiljurasz@gmail.com (J.J.); bartosz.kazmierczak@pwr.edu.pl (B.K.)
2   School of Environmental Science and Engineering, Tianjin University, Tianjin 300350, China;
    wdzheng@tju.edu.cn
*   Correspondence: piotr.jadwiszczak@pwr.edu.pl (P.J.); elzbieta.niemierka@pwr.edu.pl (E.N.)

**Abstract:** Heating and cooling sectors contribute to approximately 50% of energy consumption in the European Union. Considering the fact that heating is mostly based on fossil fuels, it is then evident that its decarbonization is one of the crucial tasks for achieving climate change prevention goals. At the same time, electricity sectors across the globe are undergoing a rapid transformation in order to accommodate the growing capacities of non-dispatchable solar and wind generators. One of the proposed solutions to achieve heating sector decarbonization and non-dispatchable generators power system integration is sector coupling, where heat pumps are perceived as a perfect fit. Air source heat pumps enable a rapid improvement in local air quality by replacing conventional heating sources, but at the same time, they put additional stress on the power system. The emissions associated with heat pump operation are a combination of power system energy mix, weather conditions and heat pump technology. Taking the above into consideration, this paper presents an approach to estimate which of the mentioned factors has the highest impact on heat pump emissions. Due to low air quality during the heating season, undergoing a power system transformation (with a relatively low share of renewables) in a case study located in Poland is considered. The results of the conducted analysis revealed that for a scenario where an air-to-water (A/W) heat pump is supposed to cover space and domestic hot water load, its CO$_2$ emissions are shaped by country-specific energy mix (55.2%), heat pump technology (coefficient of performance) (33.9%) and, to a lesser extent, by changing climate (10.9%). The outcome of this paper can be used by policy makers in designing decarbonization strategies and funding distribution.

**Keywords:** heat pump; CO$_2$ emission; power to heat; emission driving forces

## 1. Introduction

Since the heating and cooling sectors contribute to approximately 50% [1,2] of the European Union (EU) energy consumption, it is commonly recognized that they have to be decarbonized in order to meet climate change prevention goals. This is particularly important considering the fact that 75% [3] of heating and cooling comes from fossil fuels, of which 69% [4] are imported.

The advent of large-scale weather- and climate-driven renewable energy sources—in brief, variable renewable energy sources (VRESs)—has significantly altered the energy landscape. The most dramatic changes are being observed on the supply side, but the demand side is also evolving due to the emergence of new loads such as electrical vehicles. Consequently, national power systems are undergoing a process of rapid and mandatory transformation. At the same time, the non-dispatchable characteristic of solar and wind sources makes their integration into the power system a challenging and multifaceted task [5].

In recent years, multiple approaches have been proposed in the literature to facilitate their integration [6], such as spatial interconnection of geographically dispersed variable

sources; complementary operation of non-dispatchable and dispatchable generators; demand response and flexible load shifting; electricity storage; oversizing peak renewable capacity; vehicle to grid; forecasting the weather to improve power system dispatch.

Many of them are already being implemented. From the above, of particular interest for this paper is the concept of power to heat conversion. A natural sector coupling [7] where, in theory, available and cheap renewable energy is absorbed by various devices (mostly resistive heaters and heat pumps) to supply the heat demand. Conversion of renewable power to heat not only facilitates VRES integration and smoothens the market and power system operation but also reduces conventional fuels' consumption, which otherwise would have to be used to cover the heat demand.

The decarbonization of the heat sector via heat pumps undoubtedly has many benefits. From the perspective of countries which experience very low air quality (as shown in Figure 1), heat pumps can significantly contribute to solving this problem. The distributed emissions of pollutants (individual fossil fuel-based space heating) are replaced by emissions on the level of power plants. If the latter are mostly renewables or low-emission ones, the corresponding emissions from the heating sector will most likely be significantly lower.

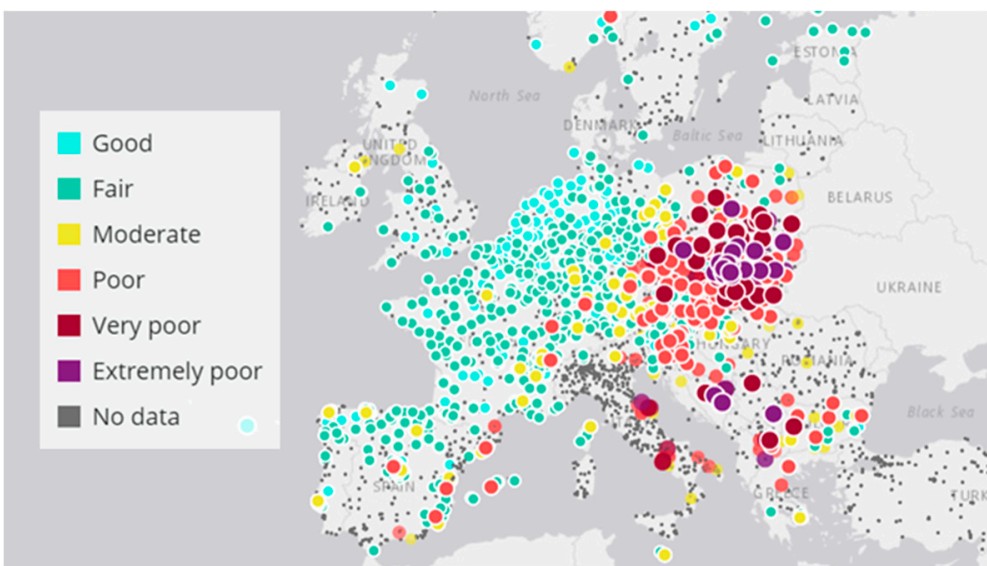

**Figure 1.** Low air quality index in Poland—carbon-based heating and power (source: [8]).

Furthermore, large-scale power stations are heavily regulated, also from the environmental point of view. Even if they emit significant amounts of $CO_2$ and other greenhouse gases (GHGs) (as in case of hard or lignite powered ones) which impact the climate, it is much easier to control the emissions of harmful substances to human beings through regulatory decisions. Taking into account the above, it is clear that heat pumps have a significant potential to improve the local air quality. However, to consider them as a clean energy source, one has to take a detailed look at the country-specific $CO_2$ emissions associated with each kWh of electricity generated. This is of particular importance in countries such as Poland in which the process of decarbonizing the power system is still in the early stages and, therefore, the majority of electricity comes from combustion of fossil fuels, as shown in Figure 2. In summary, it is obvious that despite heat pumps' ability to convert electricity into useful heat with high efficiency, the heat they generate has some associated emissions linked directly to the power system emissions.

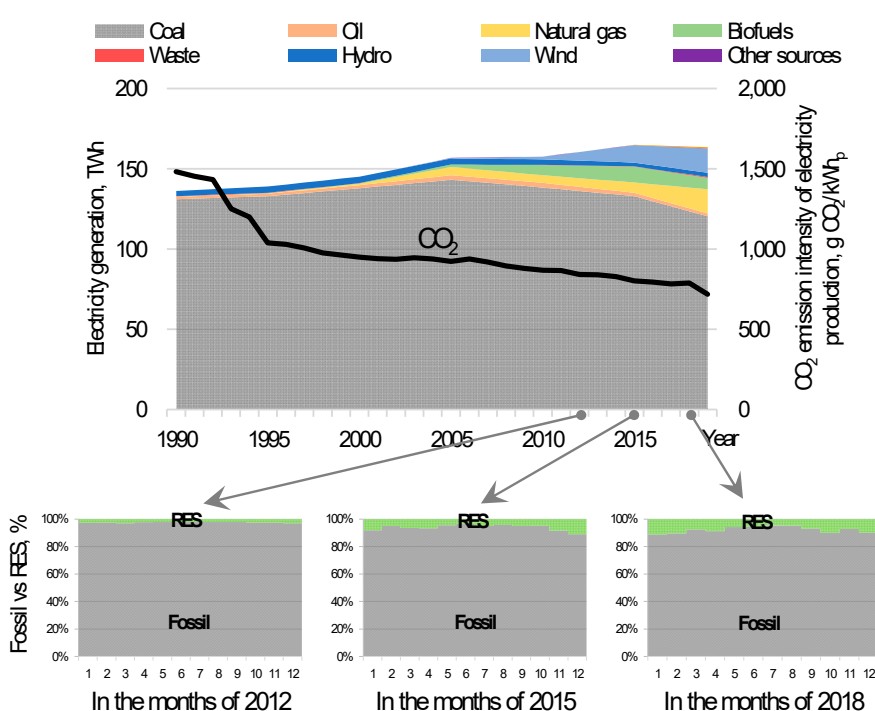

**Figure 2.** Fuel mix and $CO_2$ emission intensity of the Polish power system (data sources [9–13]).

In recent years, multiple studies have been dedicated to the analysis of heat pumps' role in shrinking the heating sector's carbon footprint. A review article by Bayer et al. presented an overview on how ground source heat pumps can contribute to reducing the greenhouse gas emissions across European countries [14]. Their findings indicate that with fully saturated markets, a reduction by 30% could be expected compared to present heating practices. In the case of Poland, the benefits of residential heat pumps in terms of GHG reduction are expected to be very low due to the high carbon intensity of the electricity generation sector. The above findings are in line with those reported by Gajewski et al., who analyzed the emissions associated with heating in various European countries [15]. Their findings highlight that the "greenness" of a heat pump depends on the country's electricity mix. Furthermore, they indicate that if fossil fuels are the main source of electricity in a given power system, then condensing gas boilers are a more ecological solution than heat pumps (again, ground source heat pumps were considered); such findings can be also found in Dzikuć and Adamczyk [16]. As a sort of a follow-up work, Sewastianik and Gajewski analyzed the emissions associated with different types of heat pumps operating in different cities representing typical climate conditions in Poland [17]. Their findings indicate that air-to-water heat pumps are, energy-wise, a non-profitable investment from the perspective of the Polish energy mix. Some contrary findings have, however, been reported by Nemś et al. when heat pumps (air-to-water and ground source) were analyzed from economic and environmental perspectives for a greenhouse heating system [18]. Nemś et al. reported that both types of heat pumps are economically viable and contribute to significant reductions in environment pollution. It has to be noted that the reference heating system was based on a coal boiler. When a condensing gas boiler is used as a reference heat source, according to Gajewski et al., the air-to-water heat pump is neither an economically viable nor environmentally beneficial solution [19]. Another more comprehensive (considering more heat sources) study by Hałacz et al. revealed that a condensing boiler with natural ventilation in case of single-family houses is, pollutant emission-wise, the most favorable solution [20]. The above-mentioned studies support the argument that the country-specific energy mix has a significant impact on the "greenness" of heat pumps. The currently ongoing changes in the structure of the Polish energy mix

indicate a promising trend (decarbonization), but the question emerges of whether this will be the only factor that will shape the "greenness" of heat pumps in the Polish context.

As shown earlier in Figure 2, the structure of the energy mix in Poland is fairly stable during the year, with a clear and dominant share of large-scale conventional fossil fuel (mostly lignite and hard coal) power stations and a marginal contribution of VRESs. At the same time, the majority of heat pumps in Poland operate as small-scale, decentralized units powered by a centralized power station. The most common are air-to-water heat pumps, which provide domestic hot water and space heating (Figure 3). Their high popularity is mostly due to their low investment costs, fast installation process and adaptability to local conditions—which makes them a perfect fit in urban environments.

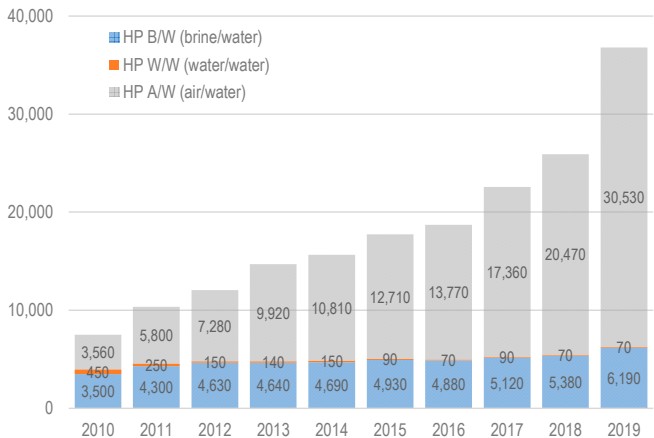

**Figure 3.** Heat pump sale in Poland by type of heat pump (source: [21]).

To analyze the environmental impact of air source heat pumps in terms of $CO_2$ emissions per kWh of heat delivered, one has to analyze which factors determine those emissions. Awareness of their role and long-term change will not only enable a better understanding of the role of heat pumps (HPs) in heat and power systems' transformation but will also facilitate the process of developing decarbonization actions/policies, which, in the long term, should enable a carbon-neutral energy sector.

In the Polish context, the electrification of the heating sector via air-to-water (A/W) heat pumps has two sides, a positive and a negative one. In general, HPs are able to eliminate local emissions and lead to air quality improvement. However, at the same time, their electricity demand puts additional stress on the power system operation. This is particularly important in countries such as Poland with a winter-peaking power system, which is dominated by conventional, highly emitting generators.

Considering the above, the objective of this work can be formulated in the form of the following research questions:

- Do heat pumps in the context of the present Polish power system support the process of heating sector decarbonization and achieving EU targets?
- What is the main factor shaping heat pump-derived heat emissivity? Is it climate change, technology progress or the power system energy mix?

## 2. Data and Methods

Considering the objective of this work, the process of heating sector decarbonization by using heat pumps in the context of the present Polish power system has been considered. The analysis included the determination of trends resulting from the long-term variability of $CO_2$ emissions accompanying the operation of air-to-water heat pumps in common applications—for heating (supply standard, 55 °C, and low-temperature sink, 35 °C) and for domestic hot water in the Polish climatic conditions and power system structure. The analysis was carried out for the city of Wrocław located in Poland in Central Europe (51°06′00″ N, 17°01′59″ E). The research covered a period of 25 years—from 1995 to 2019—

and included three driving forces influencing heat pumps' $CO_2$ emission intensity: climate change, HP technology development and the carbon emissivity of the power system.

### 2.1. Alternative Scenarios

The research was based on proposed alternative scenarios' methodology to investigate and illustrate the impact of the heat pumps application on 1995–2019 long-term decarbonization in Poland. Four characteristic scenarios, described below, have been distinguished and determined.

■ Scenario S0 presents the semi-real 1995–2019 heat pump $CO_2$ emission intensity ($EI_{HP}$) calculated based on three main factors: inventoried $CO_2$ emission intensity of electricity production ($EI_E$), historical coefficient of performance (COP) and global warming represented by ambient air temperature ($t_e$). The S0 scenario assessed the synergy of climate change, historical technology improvement and power sector decarbonization and estimated the potential benefits.

■ In scenario S1, the power sector's $CO_2$ emission intensity ($EI_E$) was fixed at the 1995 level. The 1995–2019 tendency of heat pumps' $CO_2$ emission intensity ($EI_{HP}$) was calculated with historical COP and recorded ambient air temperature ($t_e$) reflecting the influence of actions independent of or lack of development in power sector technology and no changes in fuel mix.

■ In scenario S2, the heat pumps' coefficient of performance (COP) was fixed at the 1995 value. The calculated $CO_2$ intensity of heat pumps reflects two different situations. First, the emission effect of a heat pump installed in 1995 operating on the recorded ambient air temperature ($t_e$) and powered by electricity with historical $CO_2$ emission intensity changes ($EI_E$). Finally, the lack of progress in heat pumps technology was simultaneously determined.

■ In scenario S3, the ambient air temperature ($t_e$) impacting the heat pump COP was fixed at the 1995 value. The $CO_2$ emission tendency followed the real 1995–2019 changes in power sector emission intensity and technological development of heat pumps. The S3 scenario demonstrates the influence of factors other than ambient air warming as a hypothetical value that could have been achieved if the goals relating to climate change had been implemented earlier. Simultaneously, it includes the consideration of only direct influence factors displaying the results of active improvement.

### 2.2. Input Datasets

The investigated $CO_2$ emission intensities of heat production by the heat pump ($EI_{HP}$) were calculated based on the main driving forces (Figure 4) collected in three datasets:

• Inventoried $CO_2$ emission intensity of electricity production $EI_E$ (1) [10];
• Mean air-to-water heat pump COP changes (2) [22];
• Historical ambient air temperature data (3) [23].

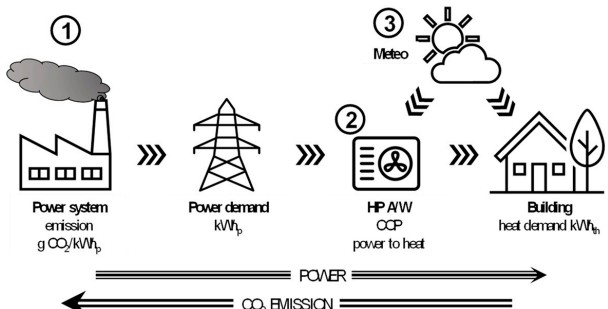

**Figure 4.** Diagram of power flows and $CO_2$ emissions of an air source heat pump in power to heat mode.

2.2.1. $CO_2$ Emission Intensity of Electricity Production $EI_E$

Emissions of greenhouse gases, especially of $CO_2$, are recorded as inventories and submitted to the United Nations Framework Convention on Climate Change (UNFCCC). In addition to the fact that they form the official data for international climate policies, they also constitute a database for the analysis of emissions due to separate classification of emissions by technical processes. Data provided by the European Environment Agency (EEA) [10] allowed to determine the average annual $CO_2$ emission intensity of electricity production ($EI_E$) expressed in grams of emitted $CO_2$ per energy unit of produced power ($g/kWh_P$). The fossil fuel-dependent power sector in Poland is characterized by an almost constant emission intensity of electricity generation, not strongly related to the day, season and weather data, as is the case with VRESs. This consistency is also visible in the Polish Power Grid monthly stepped reports [10], also presented in Figure 2. Applying the annual emissions from power generation to this methodology is, therefore, justified due to their negligible error. In a later part of the paper, the data in the annual step are used, taking into account the long-term variability and the tendency of $EI_E$ for Poland, as shown in Figure 5.

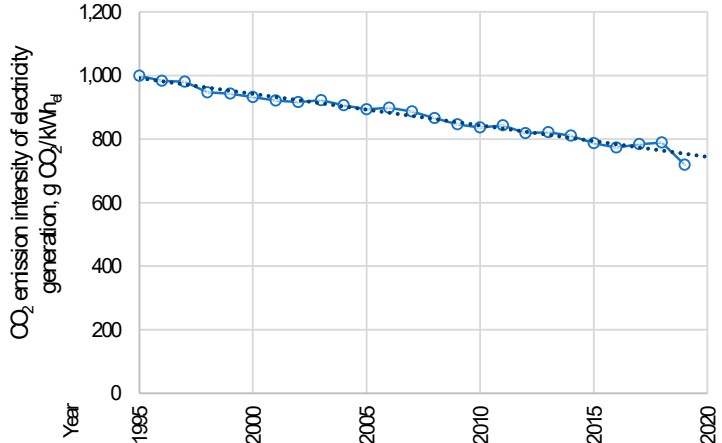

**Figure 5.** Variability of $CO_2$ emission intensity of electricity production $EI_E$ in 1995–2019 (own study based on data [10]).

2.2.2. Heat Pumps Coefficient of Performance COP

The interest in renewable energy sources and the desire to improve products have resulted in significant technological development. The evaluation of the advancement process can be analyzed on the different levels which are dominated by the estimation of multi-annual changes in heat pump coefficient of performance (COP). The increase in COP reflects a reduction in the share of driving energy required to produce the same amount of thermal energy. To estimate the intensity of emissions from heat pumps, it was necessary to determine the variation of COP among the analyzed period. The data of the annual mean COP of air-to-water heat pumps were obtained due to the products' quality monitoring [22]. The technological flow was smooth, and any possible differences between the noted and the real technical values were unnoticeable. Achieved data of annual mean COP of A/W heat pumps were reported for specific parameters—ambient air temperature equal to 2 °C and water temperature of heat sink of 35 °C, as shown in Figure 6.

In the case of heat pumps, the COP was determined using two slightly different methodologies according to EN 255 [24] and EN 14511 [25]. Due to differences in the assemblies of both methods, it was necessary to standardize the COP indicator in transitional period between 2005 and 2012. A proportional increase in the number of heat pumps compliant with the new standard was applied over these years, reaching 100% in 2012.

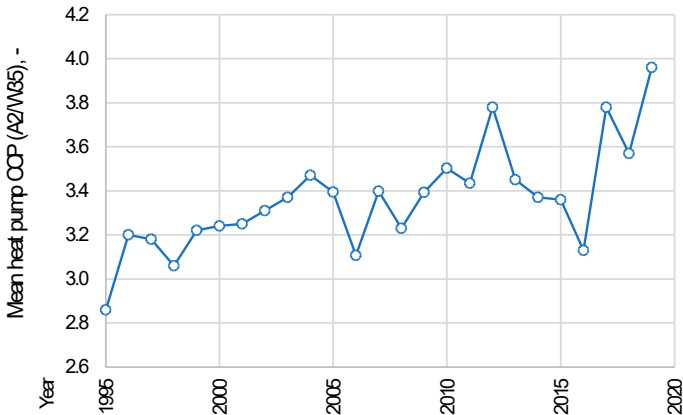

**Figure 6.** Variability of mean heat pump coefficient of performance (COP) according to EN 255 and EN 14511 in 1995–2019 (own study based on data [22]).

Conversion of the achieved dataset into continuous operating characteristics was possible due to the transformation of the annual median COP read from historical data [22] with the tendency of actual operating characteristic using proportionality ratio. Application of the characteristics of any other air-to-water heat pump would allow to obtain similar characteristics due to the homogeneity in HP technology and production conditions. The obtained dependencies allowed determination of the COP for any ambient air temperature within the operation condition range for supply of 35 and 55 °C.

### 2.2.3. Ambient Air Temperature $t_e$

The last part of the database used was a set of archive weather conditions [23]. Due to the key influence of the ambient air temperature on the achieved COP efficiency, implementing the weather data was crucial. Weather conditions can be described in an hourly or seasonal way. In the analysis of alternative scenarios, average annual air temperatures were determined on the basis of hourly data. For the calculations of domestic hot water mode, the average annual ambient air temperature was used, due to the required access to utilities throughout the year. For space heating operation mode in cold Polish climatic conditions, the average temperature for the heating season was assumed for the months October–May (inclusive) and external temperatures lower than 17 °C. Global warming and rising ambient air temperatures are visible even for the short, analyzed period of 1995–2015. Figure 7 shows the variability of the annual average air temperature (grey points) and the average temperature during the heating season (blue points). In both cases, an upward trend was noted.

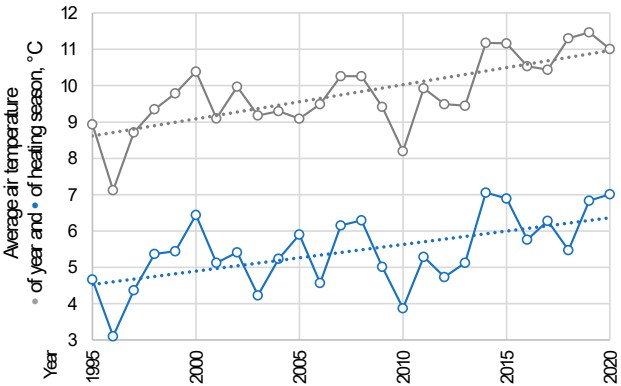

**Figure 7.** Variability of average air temperature of the year and average air temperature of the heating season in 1995–2019 (own study based on data [23]).

### 2.3. CO$_2$ Emission Intensities of Heat Pumps

The methodology of calculating the CO$_2$ emission intensity of heat production by a heat pump (EI$_{HP}$) is shown schematically in Figure 8, including input and output data.

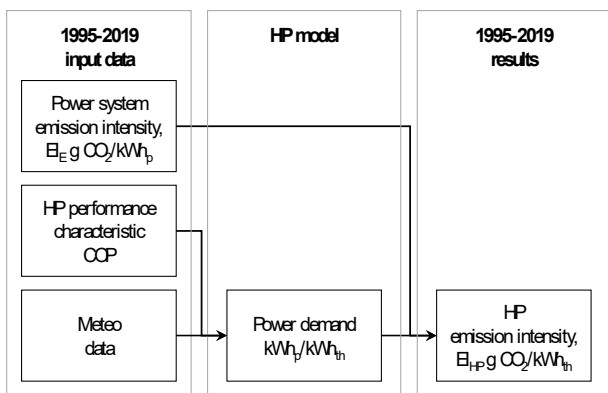

**Figure 8.** Information flow diagram for calculating the CO$_2$ emission intensities of heat pumps.

The S0 scenario consists in determining the average ambient air temperatures in each year (depending on the variant—annual or seasonal) and, based on the variability of the coefficient of performance, defines the COP for the assumed $t_e$ temperatures (Equation (1)). Then, based on the obtained COP and the CO$_2$ emission intensity of electricity production EI$_E$, the CO$_2$ emission intensity of heat production by the heat pump (EI$_{HP}$) per unit of heat produced is calculated (Equation (2)). According to the proposed alternative scenarios methodology, the calculations were made as per assumptions of the S1, S2 and S3 scenarios, including the specific driving forces fixed at the 1995 level.

$$\mathrm{COP}_{i,j} = f\left(t_{e_{i,j}}\right) \tag{1}$$

$$\mathrm{EI_{HP}}_{i,j} = \mathrm{EI_{E}}_{i,j} \cdot \frac{1}{\mathrm{COP}_{i,j}} \tag{2}$$

where:

COP—heat pump coefficient of performance;

$t_e$—average annual (domestic hot water variant) or seasonal (space heating variant) ambient air temperature, °C;

EI$_{HP}$—CO$_2$ emission intensities of heat pump per unit of produced thermal energy, g CO$_2$/kWh$_{th}$;

EI$_E$— CO$_2$ emission intensity of electricity production EI$_E$ per unit of generated power, g CO$_2$/kWh$_p$;

i—year;

j—operation mode in analyzed scenario.

## 3. Results and Discussion

This section presents and discusses the results obtained from the conducted analysis. As indicated in the earlier sections, the proposed approach for analyzing the factors shaping the CO$_2$ emissions associated with heat pump operation has been implemented on a case study from Poland.

Figure 9 shows the long-term changes in EI$_{HP}$ calculated according to four alternative scenarios, where the grey area reflects the range of available CO$_2$ emission intensities dependent on the heat sink parameters. The lines reflect common heat sinks, where solid blue represents a space heating line with a sink temperature of 55 °C, dashed blue is domestic hot water preparation (55 °C according to Polish law), and the grey, low-temperature space heating with a sink temperature of 35 °C.

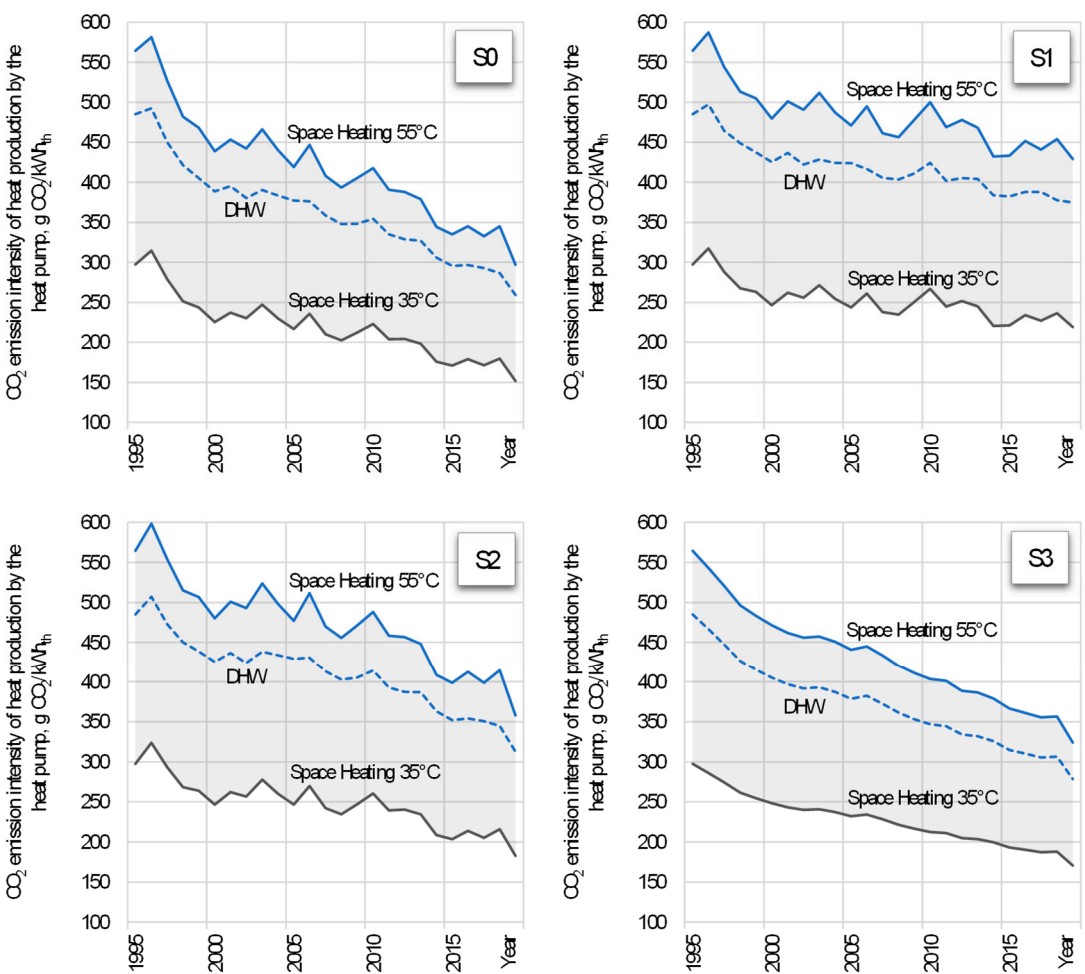

**Figure 9.** Multi-annual $CO_2$ emission intensity of heat generation by heat pump per 1 kWh$_{th}$ of produced thermal energy for different scenarios: (**S0**) semi-real; (**S1**) with fixed EI$_E$; (**S2**) with fixed COP; (**S3**) with fixed $t_e$ at the 1995 level. Solid blue line represents EI$_{HP}$ in space heating mode with a sink temperature of 55 °C, dashed blue is EI$_{HP}$ in domestic hot water (DHW) preparation mode (supply 55 °C) and the grey line is EI$_{HP}$ in a low-temperature space heating system with a sink temperature of 35 °C; grey area reflects the range of available $CO_2$ emission intensities.

S0 reflects the historical emissions, EI$_{HP}$, of working HP A/W including the 1995–2019 COP development, EI$_E$ decrease and ambient temperature increase. The HP unit operating in space heating mode at the supply temperature of 35 °C has the lowest EI$_{HP}$ due to the high efficiency achieved (high COP) and, thus, the low electricity demand to produce 1 kWh$_{th}$ (grey solid line). The highest EI$_{HP}$ is characteristic for a HP operating in heating mode at 55 °C (blue solid line). Low heat source temperatures in heating season (low ambient temperatures in Poland in winter) result in low HP efficiency (low COP) and high electricity consumption to produce 1 kWh$_{th}$. A heat pump operating in domestic hot water preparation mode (supply 55 °C) reaches high EI$_{HP}$ values, but lower than for heating 55 °C, due to high-efficiency operation in the summer months when the heat source temperature is favorable for the heat pump, which affects the seasonal result.

The value and slope of the 1995–2019 long-term EI$_{HP}$ change show a favorable reduction in emissions and unfavorably high values of emissions associated with HP operation in Poland, even contrary to the idea of using heat pumps.

The S1 scenario with fixed EI$_E$ confirms the beneficial effects of heat pump technology development (COP increase) and climate change (heat source temperature increase) on EI$_{HP}$ by the generation of 1 kWh$_{th}$. The lack of advances in fossil fuel power generation technologies does not stop the long-term reduction in EI$_{HP}$. Increasing the efficiency of the heat pump (higher COP) gives a favorable result. The large difference in the slope

compared to S0 indicates a large impact of many years of $EI_E$ changes, although the Polish energy system is still based on coal combustion and has a high emissivity.

The chart of the S2 scenario presents $EI_{HP}$ lines with COP fixed at the 1995 level. The emission decrease illustrates a beneficial reduction in HP emissions despite the lack of improvements in heat pump technology. In all analyzed HP operating modes, mitigation is driven only by long-term improvements in the national fossil fuel energy system ($EI_E$) and 1995–2019 climate warming. In 25 years, the HP installed in 1995 produces fewer and fewer emissions, and its users passively support the decarbonization of energy and heating systems. They are the beneficiaries of positive changes beyond the boundaries of their house. This phenomenon confirms the thesis about the unused environmental potential of HP units in the Polish energy mix. Despite the lack of heat pump development, the $EI_{HP}$ decreases in all analyzed operating modes. HP emission mitigation is caused by the Polish power sector decarbonization and climate change.

In the climate-fixed S3 scenario, the decarbonization is driven only by technical improvements in central power plants and in individual HP units' efficiency in all operation modes. Smooth lines, free of disturbances from weather data, illustrate the consistent development in both areas and the effect of planned activities both at the national level and at the level of distributed heat sources.

In the investigated period of 1995–2019, the global efficiency of heat pumps increased significantly (Figure 6), and the mitigation actions in the Polish national energy mix were moderate. The national energy mix is still dominated by the combustion of fossil fuels, with high GHG emissions and low RES use (Figure 1).

The influence weight of the three investigated HP emission factors in all analyzed operation modes is presented in Figure 10. In each mode, the long-term decarbonization of the power system has the biggest impact on $EI_{HP}$ emission by 53.0%, 55.2% and 55.4%, respectively. This shows and confirms the high impact and potential of decarbonization actions in central power plants and RES increase in the national energy system.

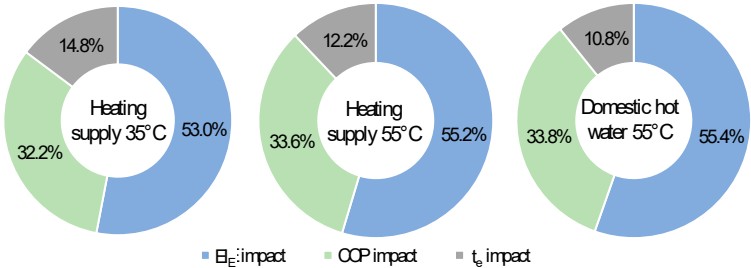

**Figure 10.** The influence weight of $EI_E$, COP and $t_e$ as emission factors of air source heat pumps.

Development in HP technology, defined by long-term COP increase, is the second force in long-term mitigation. With a share of 33%, it is an important factor in supporting the reduction in $CO_2$ emissions, with all the advantages and disadvantages of electrification of the heating sector in Poland. HP heat sink temperature and operation mode only slightly increase this share from 32.2% to 33.8%.

Observed long-term ambient air increase (1995–2019) affects the heat production efficiency in the analyzed air source heat pump technology. Apart from the sink, ambient air temperature is a factor shaping the COP. Climate change impact was weighted at 14.8%, 12.2% and 10.8% for 35 °C, 55 °C heating mode and domestic hot water (DHW) 55 °C, respectively. For the heating sector, global warming decreases the heating needs in buildings and increases the air source HP efficiency. The strongest impact concerns HP units in low-temperature heating mode (35 °C). Due to HP technology, low sink temperature systems are the greatest beneficiaries of unfavorable climate warming.

In the Polish fuel mix, all HP units powered from the national energy system are not zero or near-zero emission devices. Their operation still has a negative impact on the

environment. Despite the EU's climate obligations, the technology of power generating from fossil fuels will dominate in Poland for many years.

The high power system emission intensity ($EI_E$) paradoxically increases the HP impact on mitigation. At high system $EI_E$, HP operation, even with low COP, results in a large $CO_2$ reduction in g $CO_2/kWh_{th}$. Low $EI_E$, despite increasing the COP, weakens the HP's influence on absolute emission reduction in g $CO_2/kWh_{th}$.

The evaluated period (1995–2019) corresponds to the declared technical lifetime of a conventional heat pump. The observations made for the Polish energy sector show that the climate change mitigation potential of heat pumps has been wasted. The changes in the energy mix did not offer environmentally beneficial working conditions for heat pumps. In Poland and the EU, we are on the verge of a massive replacement of the heat pumps that were installed in 1990s. The technology has made significant progress efficiency-wise, but again, this potential might not be fully utilized in Poland. Although, there is still hope that the development of HPs will be accompanied by an increase in local, distributed renewable energy sources which will be the main or supplementary power source for them. However, such solutions might be only applicable in rural or sub-urban areas, as the densely populated urban areas usually have limited potential for VRES development that will enable a year-long HP power supply.

Proper design of a HP-based heating system should aim at supporting the decarbonization of national power and heating sectors. The improvement depends both on the public power sector and central power stations (55%) as well as individual investors and users (34%). The results have shown that the climate change in Poland is beneficial for A/W heat pumps (11%) in terms of their $CO_2$ emissions. A decreasing heat demand and increasing temperatures of heat source (here understood as ambient air temperature) during winter increase the effectiveness of the supply system. It is a paradox of benefits resulting from unfavorable changes.

## 4. Conclusions

In this paper, the environmental performance of air-to-water heat pumps has been investigated. Special attention has been paid to the impact of factors such as electrical sector energy mix, climate change and heap pump efficiency expressed as coefficient of performance. The presented method has been applied to a case study in Poland where the energy system is undergoing a rather slow process of decarbonization.

The analysis has been conducted based on annual data covering the period 1995–2019. From the obtained results, we conclude that considering the current electricity mix of the Polish energy sector, air-to-water heat pumps are unable to use their pro-environmental advantages. Heat pumps can contribute to improving local air quality but their indirect emissions are not competitive with alternative heating sources—which is in line with the available literature. This study builds upon those results and adds to the existing body of knowledge by providing evidence about how individual factors contribute to heat pumps' indirect emissions. The findings indicate that the major role is played by the power sector energy mix and its emissions, followed by technology improvement (COP of heat pumps) and, lastly, by climate change (increasing temperatures).

The work presented has its limitations as it focuses only on air-to-water heat pumps and does not take into account the cooling sector. In future works, it is important to not only consider different heat pump technologies but also investigate, in more detail, the electricity sector emissions (both in temporal and spatial scales—line dynamic hourly emissions, or local energy mix). The study could also be extended to other countries as the presented method is easily transferable, and could also include the impact of climate change on the future performance of heat pumps.

**Author Contributions:** Conceptualization, P.J. and E.N.; methodology, E.N.; investigation, P.J., J.J., B.K., E.N. and W.Z.; data curation, E.N.; writing—original draft preparation, P.J., J.J., E.N. and W.Z.; writing—review and editing, J.J. and W.Z.; visualization, P.J. and E.N. All authors have read and agreed to the published version of the manuscript.

**Funding:** This research received no external funding.

**Institutional Review Board Statement:** Not applicable.

**Informed Consent Statement:** Not applicable.

**Data Availability Statement:** The data presented in this study are available on reasonable request from the corresponding author.

**Conflicts of Interest:** The authors declare no conflict of interest.

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
