# Peer review of "Factors Shaping A/W Heat Pumps CO₂ Emissions—Evidence from Poland"

_energies, doi:10.3390/en14061576_

Round 1

Reviewer 1 Report

The paper describes the influence of the energy mix in Poland on CO2 emissions for heat pumps.
The calculations and assumptions are calculated using a very simple model. The influence of the ambient temperature and the day and night fluctuations are only vaguely discussed. The strong simplifications and the very simple model represent a major flaw in the scientific direction.
This shortcomings are described in the summary as an outlook.

REMARKS
Page 6/7 Line 218: The caption should be printed on the previous page below the image.
Page 8: Equation (1) should be given an introductory sentence in context. It is currently incoherently under the graphic 8.

Reviewer 2 Report

Review page setup including left indent and line numbers on the right.

Notations on first figure. Is not clear that the notation is EI. Due to superposition of the letters.

What about reference [0].

Note first figure as 1 and not 2.

Line 134: “polish context” correct “Polish context”.

Line 159: “scenasios” correct “scenarios”.

Line 181: “te” correct with e subscript.

Line 248: On Figure 6, vertical axis, replace “,” with “.” As decimal separator.

Line 416: Correct “presetned”.

Reviewer 3 Report

The following corrections should be done before further processing;

1- In section 1, it is required to highlight the novelty of your work in comparison with previous literature, why it is novel and what are the differences. It should be compared with at least one paper

2- The following references are recommended to cite

  • Thermodynamic Optimization of a Geothermal Power Plant with a Genetic Algorithm in Two Stages
  • A Technical analysis investigating energy sustainability utilizing reliable renewable energy sources to reduce CO2 emissions in a high potential area
  • A Technical analysis investigating energy sustainability utilizing reliable renewable energy sources to reduce CO2 emissions in a high potential area
  • Energy, Exergy, Economic, and Exergoenvironmental Analyses of a Novel Hybrid System to Produce Electricity, Cooling, and Syngas
  • A parametric study to simulate the non‐Newtonian turbulent flow in spiral tubes
  • CO2 utilization via integration of an industrial post-combustion capture process with a urea plant: Process modelling and sensitivity analysis

3- The limitations of the model should be explicitly described

4- Could you please explain pressure profile?

5- What types of heat pumps are considered in the present work?

6- Discussion section should be separated and improved?

Round 2

Reviewer 3 Report

It can be published in the current version